# RAISING is a high-performance method for identifying random transgene integration sites

Yusaku Wada[1,23], Tomoo Sato[2,3,23], Hiroo Hasegawa[4,5,23], Takahiro Matsudaira [1,23], Naganori Nao[6,7], Ariella L. G. Coler-Reilly[2,8], Tomohiko Tasaka[9], Shunsuke Yamauchi[4], Tomohiro Okagawa [10], Haruka Momose[11], Michikazu Tanio[11], Madoka Kuramitsu[11], Daisuke Sasaki[4], Nariyoshi Matsumoto[4], Naoko Yagishita[2], Junji Yamauchi[2], Natsumi Araya[2], Kenichiro Tanabe[12], Makoto Yamagishi[13], Makoto Nakashima[13], Shingo Nakahata[14], Hidekatsu Iha[15], Masao Ogata[16], Masamichi Muramatsu[17], Yoshitaka Imaizumi[18], Kaoru Uchimaru[13], Yasushi Miyazaki[18,19], Satoru Konnai[10,20], Katsunori Yanagihara[4,5], Kazuhiro Morishita[14], Toshiki Watanabe[21], Yoshihisa Yamano [2,3,24] & Masumichi Saito [17,22,24 ✉]

Both natural viral infections and therapeutic interventions using viral vectors pose significant risks of malignant transformation. Monitoring for clonal expansion of infected cells is important for detecting cancer. Here we developed a novel method of tracking clonality via the detection of transgene integration sites. RAISING (Rapid Amplification of Integration Sites without Interference by Genomic DNA contamination) is a sensitive, inexpensive alternative to established methods. Its compatibility with Sanger sequencing combined with our CLOVA (Clonality Value) software is critical for those without access to expensive high throughput sequencing. We analyzed samples from 688 individuals infected with the retrovirus HTLV-1, which causes adult T-cell leukemia/lymphoma (ATL) to model our method. We defined a clonality value identifying ATL patients with 100% sensitivity and 94.8% specificity, and our longitudinal analysis also demonstrates the usefulness of ATL risk assessment. Future studies will confirm the broad applicability of our technology, especially in the emerging gene therapy sector.

[1] Biotechnological Research Support Division, FASMAC Co., Ltd, Atsugi, Kanagawa 243-0021, Japan. [2] Department of Rare Diseases Research, Institute of Medical Science, St. Marianna University School of Medicine, Kawasaki, Kanagawa 216-8512, Japan. [3] Division of Neurology, Department of Internal Medicine, St. Marianna University School of Medicine, Kawasaki, Kanagawa 216-8511, Japan. [4] Department of Laboratory Medicine, Nagasaki University Hospital, Nagasaki 852-8501, Japan. [5] Department of Laboratory Medicine, Nagasaki University Graduate School of Biomedical Sciences, Nagasaki 852-8501, Japan. [6] Division of International Research Promotion, International Institute for Zoonosis Control, Hokkaido University, Sapporo 001-0020, Japan. [7] One Health Research Center, Hokkaido University, Sapporo 060-0818, Japan. [8] Department of Internal Medicine, Division of Bone and Mineral Diseases, Washington University School of Medicine, St. Louis, MO 63110, USA. [9] Affinity Science Corporation, Tokyo 141–0031, Japan. [10] Department of Advanced Pharmaceutics, Faculty of Veterinary Medicine, Hokkaido University, Hokkaido 060-0818, Japan. [11] Department of Safety Research on Blood and Biological Products, National Institute of Infectious Diseases, Tokyo 208-0011, Japan. [12] Pathophysiology and Bioregulation, St. Marianna University Graduate School of Medicine, Kawasaki, Kanagawa 216-8511, Japan. [13] Department of Computational Biology and Medical Sciences, Graduate School of Frontier Sciences, The University of Tokyo, Tokyo 108-8639, Japan. [14] Division of Tumor and Cellular Biochemistry, Department of Medical Sciences, University of Miyazaki, Miyazaki 889-1692, Japan. [15] Department of Microbiology, Faculty of Medicine, Oita University, Oita 879-5593, Japan. [16] Department of Hematology, Oita University Hospital, Oita 879-5593, Japan. [17] Department of Virology II, National Institute of Infectious Diseases, Tokyo 162-8640, Japan. [18] Department of Hematology, Nagasaki University Hospital, Nagasaki 852-8501, Japan. [19] Atomic Bomb Disease and Hibakusha Medicine Unit, Atomic Bomb Disease Institute, Nagasaki University, Nagasaki 852-8102, Japan. [20] Department of Disease Control, Faculty of Veterinary Medicine, Hokkaido University, Sapporo, Hokkaido 060-0818, Japan. [21] Department of Practical Management of Medical Information, St. Marianna University Graduate School of Medicine, Kawasaki, Kanagawa 216-8511, Japan. [22] Center for Emergency Preparedness and Response, National Institute of Infectious Diseases, Tokyo 162-8640, Japan. [23] These authors contributed equally: Yusaku Wada, Tomoo Sato, Hiroo Hasegawa, Takahiro Matsudaira. [24] These authors jointly supervised this work: Yoshihisa Yamano, Masumichi Saito. ✉email: saitomas@niid.go.jp

Approximately 10–15% of all human cancers are associated with viral infections[1]. A subset of viruses integrates into their host genomes, and the integration often causes malignant transformation by affecting the expression of cancer driver genes[2–4]. This phenomenon is also a known complication of retrovirus-based therapeutics. Indeed, a subset of retroviral gene therapy-treated patients with X-linked severe combined immunodeficiency developed T-cell acute lymphoblastic leukemia due to proto-oncogene activation[5,6]. Although efforts are underway to limit this problem, it is still impossible to control the viral vector integration site. Moreover, although genome editing technologies foregoing the use of viral vectors are expected to enter the gene therapy arena[7], these too have risks associated with off-target editing and unpredictable integration[8,9]. Therefore, it is critical to develop a clinically applicable method of monitoring integration sites and the clonality of transgene-integrated cells to detect early signs of cancer.

Several such methods of analyzing transgene integration events using high thoroughput sequencing (HTS) have recently been developed: namely, ligation-mediated PCR[10,11], target-capture sequencing[12,13], LAM-PCR (linear amplification-mediated PCR)[14], nrLAM-PCR (non-restrictive linear amplification-mediated PCR)[15,16], and tag-PCR[17]. Some of these methods have already been employed in clinical settings to assess the safety of retroviral and lentiviral gene therapies[14,15]. However, HTS analysis remains prohibitively expensive for the many researchers and physicians seeking a more practical solution, especially in countries where such resources are scarce[18]. In addition, there are concerns about the poor sensitivity of methods like LAM-PCR or Target-capture sequencing that utilize restriction enzymes or sonication to fragment the DNA. Since fragmentation occurs randomly, only a small fraction of fragments often include enough transgene to anneal to the primers or capture with probes and an appropriate length of host genome for further analysis, which greatly limits the sensitivity of these methods. The newer nrLAM-PCR method eliminates the restriction digest step, greatly increasing the copy number of transgene-integrated fragments. However, this method suffers from the limitations of the single-stranded linker ligation system, namely a very long 32-h run-time with low efficiency[15,16].

Previously, we developed a novel method designed to overcome the limitations mentioned above of currently available integration site analysis technologies. Known as Rapid Amplification of Integration Sites (RAIS), our protocol substituted a polyA-tailing step for the single-stranded linker ligation system, shortening the run-time from 32 to 4 h and increasing the sensitivity 100-fold[19]. While RAIS represented a major improvement over nrLAM-PCR, uptake of the technology was still limited by the high cost of biotinylated primers and magnetic streptavidin beads. Another drawback was the requirement for two ssDNA purification steps, which decreased sensitivity. With these issues in mind, we devised the new method described herein, known as Rapid Amplification of Integration Sites without Interference by Genomic DNA contamination (RAISING). Forgoing the use of biotinylated primers and magnetic beads, we invented a novel procedure using poly-AG-tailing and thermomodulation to yield short, uniform transgene-integrated fragments for amplification. Thus, we evolved our technology to produce a highly cost-effective and sensitive method that is practical for routine clinical testing and basic science research.

Here we describe the development of RAISING and demonstrate the clinical utility of our technology using the retrovirus human T-cell leukemia virus type-1 (HTLV-1) as a model[20]. HTLV-1 infects at least 5–10 million worldwide, including many in developing countries[21]. This virus can trigger aggressive cancer, adult T-cell leukemia/lymphoma (ATL), as well as a debilitating neuroinflammatory disease, HTLV-1-associated myelopathy/tropical spastic paraparesis (HAM/TSP)[22–24]. Since ATL has an extremely high case fatality rate, it is paramount to detect cancer and begin treatment as early as possible; those at high risk of developing ATL must be identified and closely monitored. Screening for patients with high HTLV-1 proviral load in the peripheral blood is reportedly useful but certainly not specific[13,25]. However, studies using HTS technology have revealed a more specific metric for ATL risk: the oligoclonality index (OCI), which quantifies the clonality of HTLV-1-infected cells, combined with the detection of somatic mutations[11,13,26]. Here we use RAISING to rapidly analyze more than 700 samples from HTLV-1-infected patients and define our metric for ATL risk assessment, demonstrating the utility of RAISING for clinically applicable integration site analysis.

## Results and discussion

**Development of RAISING.** Here we describe the development of our novel method, RAISING, comprising six steps to achieve amplification of transgene-integrated fragments within only 3.5 h, plus an additional step for Sanger sequencing or HTS library preparation (Fig. 1a, see also protocol for RAISING in Supplementary information).

*Step 1: Single-stranded DNA (ssDNA) synthesis.* This step is critical for increasing the sensitivity of RAISING by synthesizing the ssDNA of transgene-integrated fragments using a single transgene-specific primer, F1. To determine factors influencing the sensitivity, we applied RAISING to two different sources of genomic DNA as the templates for ssDNA synthesis: freshly isolated DNA and fragmented DNA with or without RNase A treatment. As expected, we found that the sensitivity of RAISING was decreased with the fragmented and RNase A-untreated genomic DNA (Supplementary Fig. 1a). In addition, we found that the sensitivity also positively correlated with the amount of genomic DNA used in this step (Supplementary Fig. 1b). We also determined that the length of RAISING final products depended on both the quality of genomic DNA and the ssDNA synthetic time (Supplementary Fig. 1a, c). Although the copy number of ssDNA can be increased with the cycle number used in the synthesis (Supplementary Fig. 1d), we decided on 25 cycles to avoid nonspecific amplification (Supplementary Fig. 1e). We could detect the appropriate length of amplified DNA with high sensitivity using KOD-Plus-Neo (see Fig. 2a) but not OneTaq and Q5 PCR enzymes (Supplementary Fig. 1f).

*Step 2: Column purification.* Here, we used the Monarch PCR and DNA cleanup kit to eliminate larger genomic DNA fragments (>10 kb) as well as the leftover F1 primer while preserving the ssDNA as well as heat-fragmented smaller genomic DNA (<10 kb, later referred to as contaminated genomic DNA). Importantly for extracting ssDNA, this kit requires a low elution volume with low dead volumes (Supplementary Fig. 2).

*Step 3: PolyAG-tailing.* The ssDNA and contaminated genomic DNA isolated in the previous step underwent polyAG-tailing instead of the polyA-tailing employed in our older RAIS procedure. This is the critical step where the present method diverges from our previous method. By performing polyAG-tailing in this step, synthesis of the adaptor primer 1 (ADP1) sequence, a part of the oligo-dT-adaptor primer 1 (oligo-dT-ADP1) at the 3′-end of the ssDNA and contaminated genomic DNA, is inhibited in steps 4 and 5. The inhibition sequentially blocks oligo-dT-ADP1-mediated amplification of contaminated genomic DNA as seen in amplicons with polyA-tailing (Supplementary Fig. 3a). Thus,

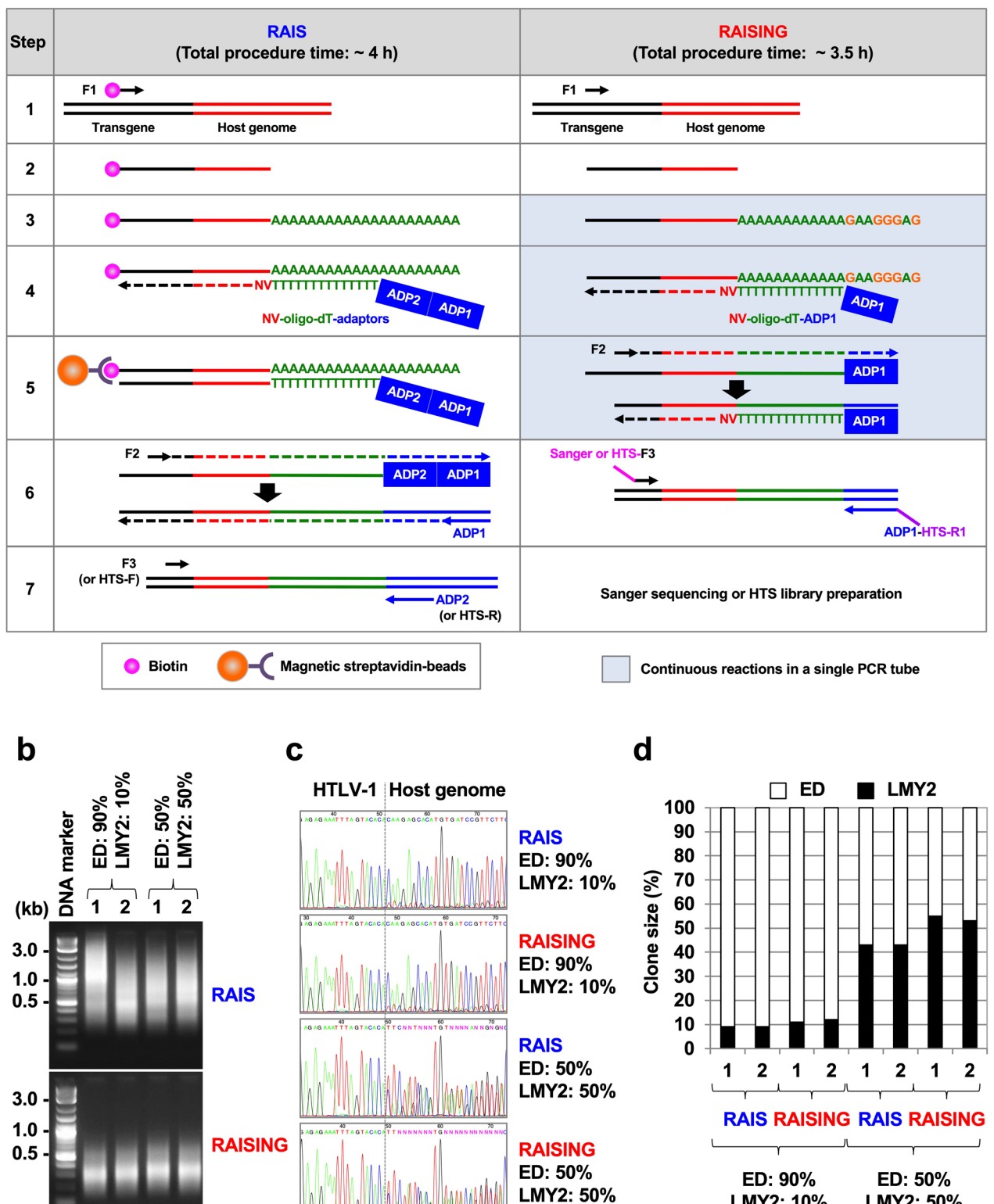

unlike RAIS, RAISING does not require a biotinylated primer and magnetic streptavidin beads for eliminating contaminated genomic DNA.

*Step 4: Double-stranded DNA synthesis.* In this step, we modulated the conditions under which oligo-dT anneals to the polyA tails to produce dsDNA (Supplementary Fig. 3b and Supplementary Data 1). In preparation for those using HTS in the final step, we elucidated which oligo-dT-ADP1 produced the most uniform amplicon lengths. We determined that NV-oligo-dT-ADP1, which includes VN (V = A or G or C, N = A or G or C or T) at the 3′-end, performed better than V-oligo-dT-ADP1 (V only) or oligo-dT-ADP1 (neither V nor N). We also tested the results using Sanger sequencing and found that in this case, all three produced similar

**Fig. 1 Characterization of Rapid Amplification of Integration Site (RAIS) and RAIS without Interference by Genomic DNA contamination (RAISING).** **a** Schematic representation of RAIS and RAISING. Step 1: Single-stranded DNA (ssDNA) synthesis; Step 2: Column purification of ssDNA; Step 3: polyA-tailing and polyAG-tailing of ssDNA in RAIS and RAISING, respectively; Step 4: Double-stranded DNA (dsDNA) synthesis; Step 5: DNA purification of dsDNA with magnetic streptavidin-beads in RAIS and first PCR in RAISING; Step 6: First PCR in RAIS and second PCR in RAISING; Step 7: Second PCR in RAIS and Sanger sequencing or high throughput sequencing (HTS) library preparation in RAISING **b–d**, Linear-amplification, accuracy, and consistency between RAIS and RAISING. Genomic DNA of LMY2 and ED cell lines, each harboring a single HTLV-1 integration site, were mixed at the indicated percentages, and these mixed samples were processed using both RAIS and RAISING. **b** Products were visualized by electrophoresis on 2% agarose gels. **c** Products were analyzed using Sanger sequencing, with dotted lines indicating the position of the HTLV-1 integration sites. **d** Products were analyzed using HTS to measure clone size. Data from two independent experiments (1 and 2) are shown.

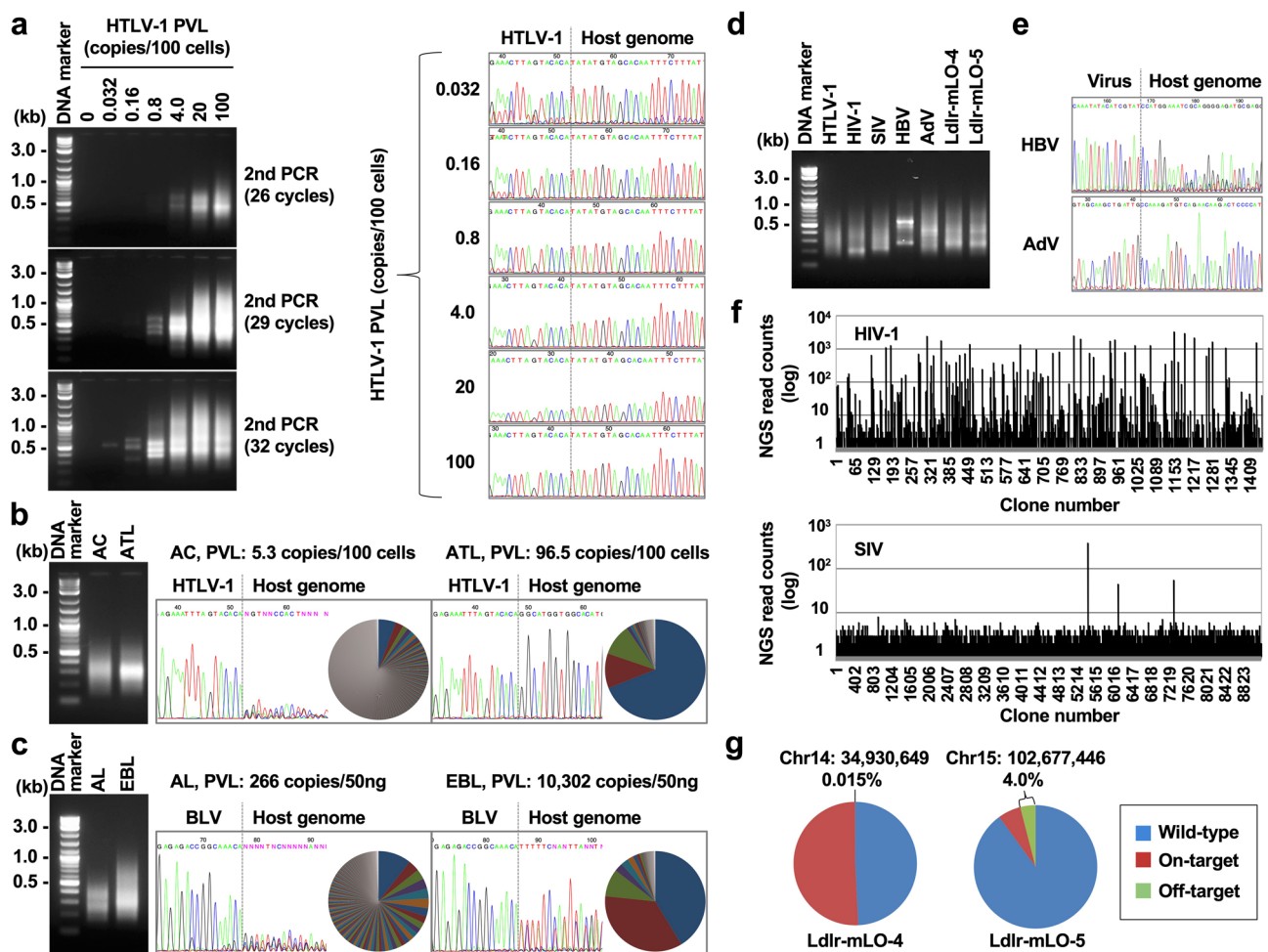

**Fig. 2 Performance of Rapid Amplification of Integration Site without Interference by Genomic DNA contamination (RAISING).** All RAISING products were visualized by electrophoresis on 2% agarose gels. Dotted lines in Sanger sequencing spectra indicate the position of transgene integration sites. **a** The sensitivity of RAISING was assessed by serially diluting TL-Om1 genomic DNA (an HTLV-1-infected cell line harboring a single copy of HTLV-1) with Jurkat genomic DNA (an HTLV-1 negative cell line). Even extremely diluted samples (resulting in PVL as low as 0.032) could be detected by increasing the cycles in the second PCR of RAISING (left panel). Sanger sequencing analysis of RAISING products confirmed that the same integration site was identified in every dilution (right panel). PVL proviral load. **b**, **c** Clonality analyses of infected samples using Sanger sequencing (spectra) vs. high throughput sequencing (HTS, pie charts) showed similar results. **b** HTLV-1 clonality analysis of an asymptomatic carrier (AC) and an adult T-cell leukemia/lymphoma (ATL) patient. **c** Bovine leukemia virus (BLV) clonality analysis of an aleukemic (AL) cow and a cow with enzootic bovine leukosis (EBL). **d** Successful amplification of various transgene-integrated fragments (HTLV-1; HIV-1 human immunodeficiency virus, SIV simian immunodeficiency virus, HBV hepatitis B virus, AdV adenovirus, and low-density lipoprotein receptor knock-in mice Ldlr-mLO-4 and Ldlr-mLO-5) with RAISING. **e** HBV and AdV integration sites. **f** Clonality analysis of HIV-1 and SIV-infected cells with HTS analysis. HTS read counts indicate the size of each clone. **g** RAISING with HTS analysis was used to identify on- and off-target effects in Ldlr-mLO-4 and Ldlr-mLO-5 knock-in mice established by genome editing technology. Chr, chromosome.

results, meaning those using Sanger sequencing can choose any of the three oligo-dT options. Next, we selected a specific temperature for the pre-denaturation step to avoid amplifying genomic DNA contaminants. Without ever heating the reaction to the usual high temperatures (around 95 °C) used for denaturation, we employed a 5 min 65 °C step followed by an annealing phase using a step-down

cycle from 64 to 52 °C. In this way, we could selectively target the ssDNA produced using the NV-oligo-dT-ADP1 primer without reaching the temperatures that would denature genomic DNA contaminants. Finally, to allow steps 3–5 to proceed sequentially in a single tube, we selected a PCR enzyme that would function efficiently among the reagents used in step 3 for polyAG-tailing. The

Q5 enzyme performed the best under these conditions (Supplementary Fig. 3c). The polyAG-tailing is exchangeable to polyTG-tailing if the transgene has a polyA sequence downstream of the specific F1 primer (Supplementary Fig. 3d).

*Step 5: First PCR.* Here, we perform the critical PCR step, introducing a transgene-specific F2 primer as the forward primer and using full-length NV-oligo-dT-ADP1 as the reverse primer. It should be noted that the ssDNA produced in step 1 is no longer the template here; because the F2 primer anneals only to the copy of the transgene produced in the previous step, it can be said that "template switching" has occurred. Thus, synthesis from the F2 forward primer produces the binding site for the full-length NV-oligo-dT-ADP1 reverse primer. Importantly, we designed the primers to bind at the relatively high annealing temperature of 68 °C; a lower annealing temperature would significantly increase the duration of this step, slowing the overall run-time of our method. This step, which employs both a forward and a reverse primer, is essential for ensuring the high specificity of RAISING. The method is also highly sensitive due to the unique design allowing steps 3–5 to proceed uninterrupted in a single PCR tube.

*Step 6: Second PCR.* We prepared a 1:200 dilution of the first PCR reaction to avoid nonspecific amplification and used two primers (Sanger-/HTS-F3 and ADP1-HTS-R1) that include half of the adaptor sequence corresponding to the Illumina high throughput sequencing (HTS) library. Transgene integration sites can be analyzed by Sanger sequencing using the amplicons from this step.

*Additional step: HTS library preparation.* The amplicons in step 6 can be re-amplified with two primers that contain 8-nucleotide sequencing indexes and the remainder of the Illumina adaptor sequence.

**Performance of RAISING.** After each RAISING step was fully evaluated, we compared the performance of RAIS and RAISING by assessing samples containing different percentages of two HTLV-1-infected cell lines (ED and LMY2)[27,28], each harboring a single integration site (Fig. 1b–d, and Supplementary Fig. 4a, b). As expected, the amplicons from RAISING exhibited a superior (denser) pattern compared to those from RAIS (Fig. 1b). With both RAIS and RAISING, we observed consistent Sanger sequence spectra patterns, clone size differing from the expected result by less than ±10%, and consistent between experimental replicates within ±2% (Fig. 1c, d). These results clearly indicate that while both RAIS and RAISING are effective, unbiased methods, RAISING is better suited to produce the final amplicon length amenable to subsequent HTS analysis.

Each method was tested on samples where genomic DNA from the HTLV-1-infected cell line TL-Om1 was serially diluted with genomic DNA from the HTLV-1-negative Jurkat cell line to compare the sensitivity limits of RAISING and RAIS[19]. While RAIS could not be used to amplify integrated fragments in samples with a proviral load (PVL) <0.16%, the limit for RAISING was PVL 0.032%, indicating RAISING achieved a five-fold higher sensitivity compared to RAIS (Fig. 2a). We also confirmed the effectiveness of cell-direct RAISING on samples of HTLV-1-infected cell lines (KK1 and SLB-1) harboring multiple integration sites (Supplementary Fig. 5a–c).

Consistent with previous results achieved using RAIS[19], RAISING discriminated between non-malignant and malignant samples in both HTLV-1-infected human specimens and bovine leukemia virus (BLV)-infected cattle specimens successfully, regardless of whether Sanger sequencing or HTS was used for

analysis (Fig. 2b, c, and Supplementary Data 2). By simply changing the transgene-specific primer sets, we could use the same RAISING method to amplify transgene-integrated fragments from a variety of virus-infected and genome-edited samples (HTLV-1; HIV-1, human immunodeficiency virus; SIV, simian immunodeficiency virus; HBV, hepatitis B virus; AdV, adenovirus; and low-density lipoprotein receptor knock-in mice Ldlr-mLO-4 and Ldlr-mLO-5) (Fig. 2d). When analyzing monoclonal expanded cells, we could identify the integration site using Sanger sequencing (Fig. 2e); for the analysis of polyclonally expanded cells, we could detect both the integration site and the size of each clone using HTS analysis (Fig. 2f). Importantly, RAISING with HTS analysis also successfully discriminated between on- and off-targets integrating the Loxp donor sequence into an unpredictable position in the genome-edited Ldlr-knock-in mice (Fig. 2g). Collectively, these results demonstrate that RAISING is a high-performance method for characterizing multiple-transgene integration events.

**Practical application of RAISING-CLOVA in a clinical field.** Quantifying the clonality of HTLV-1-infected cells in patients is essential for assessing their risk of developing aggressive cancer ATL. In this study, we developed our software, known as CLOVA (Clonality value), that can automatically provide a clonality value (Cv) simply by uploading Sanger sequencing data (ab1 file), entering the proviral sequence (up to 20 bp adjacent to the host genome sequence in 5′–3′ orientation), and selecting the desired nucleotide length for analysis (Supplementary Fig. 6a, b)[19].

We demonstrated that visual Sanger sequence spectra could be converted to a quantitative HTLV-1 clonality value. Theoretically, the total signal peak area for 20 nucleotides of the HTLV-1 sequence and 20 nucleotides of the host genome sequence should be identical in a monoclonal sample. In other words, there is a 1:1 ratio between the host genome and HTLV-1 spectral areas, which we classify as a Cv of 1.00. With polyclonal samples, increasing the number of clones reduces the intensity of the host genome signal, and thereby the host genome spectral area, lowering this ratio and yielding Cv < 1.00 (Supplementary Fig. 6c). Practically, slight differences in intensity between different nucleotide sequences exert a minor influence on the Cv. For example, occasionally, a sample may yield Cv > 1.00, which should be interpreted as Cv = 1.00, a monoclonal sample. We also showed that Cv accurately reflected the size of the dominant clone (first clone), as measured using HTS analysis (Supplementary Fig. 7a, b). We have previously shown that multiclonal expansion of malignant cells occurs in ~30% of ATL patients[19]. Therefore, we incorporated a function into CLOVA that provides nucleotide sequences for both the first and second clones. Subsequently, these sequences can be searched on BLAST homology to confirm that two integration sites have been accurately identified (Supplementary Fig. 8a, b).

To measure the precision of RAISING-CLOVA, we performed five independent analyses on four samples from HTLV-1-infected subjects with differing Cv values (sample A, 0.88; sample B, 0.57; sample C, 0.40; sample D, 0.12), and we found the variation in Cv was less than ±0.03 in all samples (Supplementary Fig. 9). Similarly, two different laboratories measured the clonality of the same HTLV-1-infected samples using the RAISING-CLOVA method with a high interrater agreement (Supplementary Fig. 10, n = 62). Finally, we tested whether RAISING-CLOVA could be used to measure clonality when applied directly to whole blood samples without isolating peripheral blood mononuclear cells (PBMCs). We tested RAISING-CLOVA on whole blood samples and PBMCs from the same patients and determined that the performance was similar (Supplementary Fig. 11, n = 51).

We began investigating the clinical utility of RAISING-CLOVA by assessing the suitability of Cv for discriminating between samples from patients with and without the HTLV-1-associated cancer ATL. Since HTLV-1 proviral load (PVL) has long been considered an important marker for the development of ATL[25,29], we directly compared the performance accuracy of Cv and PVL. We determined the Cv of specimens from asymptomatic carriers (AC, $n = 201$), patients with HAM/TSP ($n = 223$), and ATL patients ($n = 286$). Of these 710 subjects, blood samples suitable for measuring HTLV-1 proviral load (PVL) were available for 688 (Supplementary Data 3). As reported previously, patients with HAM/TSP and ATL exhibited significantly higher PVL than ACs (Fig. 3a)[30,31]. Within the ATL group, patients with chronic or acute subtypes exhibited significantly higher PVL than those with smoldering subtypes (Supplementary Fig. 12a). On the other hand, Cv in patients with HAM/TSP was as low as that of ACs, and patients with ATL exhibited significantly higher Cv than both ACs and those with HAM/TSP (Fig. 3b). Within ATL subtypes, Cv of the smoldering type was significantly lower (Supplementary Fig. 12b), supporting the clonal progression model for ATL that we proposed previously[32]. Upon analyzing receiver operating characteristics (ROC), we determined that Cv analysis was more effective for discriminating ATL from both AC and HAM/TSP than PVL analysis (Cv: AUC 0.996, PVL: AUC 0.941, Supplementary Fig. 13a, b). We also compared our results to the oligoclonality index (OCI) by Gillet NA et al. [11], another metric for quantifying HTLV-1 clonality. We found a correlation between Cv and OCI (Supplementary Fig. 14) and similar clonality patterns among AC, HAM/TSP, and ATL. These suggest that Cv from RAISING-CLOVA can be an alternative to OCI.

To assess the diagnostic value of our technology, we proceeded to define cut-off values for distinguishing subjects with and without ATL using Cv and PVL (Fig. 3c). As a preliminary step, we presupposed that subjects with a PVL <0.5% might be classified as non-ATL without measuring Cv. We have confirmed that RAISING-CLOVA can accurately measure Cv when the PVL is at least 0.5%, but not lower (Supplementary Fig. 15). Though this may be considered a technical limitation, it is well established that ATL risk increases with higher PVL, so it may not be useful to measure Cv when PVL is below this very low threshold. Indeed, in our study, we report no ATL patients with PVL <0.5%. Next, among subjects with a PVL of at least 0.5%, we determined a Cv cut-off value of 0.48, which could distinguish all subjects with and without ATL with 100% sensitivity and 94.8% specificity. For comparison, PVL ≥ 4% has been previously proposed as a diagnostic test[25], and this cut-off value would produce only 96.2% sensitivity and 62.3% specificity.

We then took the first steps towards classifying a potential ATL risk by Cv. We decided that subjects with Cv above the cut-off as mentioned earlier of 0.48 would be classified as zone 3 and ATL zone (Table 1). We confirmed that even non-ATL subjects above this cut-off possessed a large dominant clone (≥~40% of the total infected cells), which is highly suggestive of ATL complication as well as malignant expansion. To find a cut-off for classifying the remaining subjects into the other two zones, we analyzed the distribution of Cv values among subjects without ATL (Fig. 3d, $n = 364$). We employed a common method using the derivative function with intervals of 0.05 to identify the inflection point Cv 0.25. The vast majority (86%) of non-ATL subjects in our study thus fell into a typical non-ATL category (zone 1), with Cv < 0.25. In addition, the dominant clone size in this zone was all relatively small (<~13%), suggesting a low risk of progression to ATL.

Of course, any risk assessment analysis using only cross-sectional data should be interpreted with caution, and it is important to validate our proposed Cv cut-off values in a longitudinal study. Therefore, we conducted a retrospective longitudinal analysis using available samples from 15 progressors to ATL and 130 non-progressors (Fig. 4 and Supplementary Data 4). In this analysis, we assessed the prognostic value of Cv compared with established markers such as PVL and soluble IL-2 receptor (sIL-2R) (Fig. 4a). ROC analysis was performed using available data in a time point between one month and one year before either ATL onset or final visit of non-progressors (progressors to ATL, median 7.0 months ago; non-progressor, median 6.2 months ago). The results demonstrated that Cv was the most effective marker to distinguish between progressors and non-progressors (Fig. 4b, Cv: AUC 0.880, PVL: AUC 0.738, sIL-2R: 0.782). We determined Cv 0.50 as a cut-off value that rendered enough specificity (≥95%) to identify ATL high-risk AC and HAM/TSP patients (Table 2). The cut-off values for PVL and sIL-2R with similar specificity were 13.5% and 1260 U/mL, respectively, but their sensitivities were much lower than that of Cv. Likewise, we determined Cv 0.24 as a cut-off value that rendered enough sensitivity (≥80%) to identify ATL middle-risk patients. The cut-off values for PVL and sIL-2R with similar sensitivity were 5.0% and 479 U/mL, respectively, but their specificities were much lower than that of Cv. As a result, the longitudinal analysis provided quite similar cut-off values as obtained from cross-sectional data. In this study, we found that clonal expansion of (pre-) leukemic cells was detectable by increasing Cv months or even years. Therefore, these suggest that rising Cv may be an effective early warning sign for clinicians monitoring these patients. Finally, the dramatic decrease in Cv after one of the ATL patients was treated suggests that RAISING-CLOVA may also be a useful tool for assessing the effectiveness of ATL therapeutic agents (Supplementary Fig. 16 and Supplementary Data 4). We used HTS analysis to confirm that the sharp decrease in Cv did indeed correspond to eradicating the dominant clone in this patient. Collectively, these results suggest it may be beneficial to monitor patients with HTLV-1 using RAISING-CLOVA to detect the earliest signs of progression to ATL. Once high-risk individuals are identified, we recommend supplementing Cv analysis by pursuing mutational profiling of genes mutated in ATL frequently[13,26,33].

Thus, we performed a comprehensive HTLV-1 clonality analysis with RAISING-CLOVA and developed a method for early detection and risk assessment for progression to ATL among HTLV-1-infected patients. However, there are several limitations in this study that should be noted. Firstly, we found that ~2% of our ATL specimens ($n = 5$) carried a dominant clone with a variant of the provirus known to lack the 3' long terminal repeat where HTLV-1-specific F2 and F3 primers bind[12,34]. Unsurprisingly, RAISING failed to the HTLV-1-integrated fragment of the dominant clone in these cases. Therefore, it is important to keep this limitation in mind and use another method such as southern blot analysis or target-capture sequencing to determine clonality in these patients[13,19]. Secondly, regarding the ATL risk assessment method developed herein, we only used peripheral blood samples of HTLV-1-infected individuals, which may only be accurate for predicting the leukemia subtypes of ATL. For example, in a patient who developed the lymphoma subtype, we found that the Cv in peripheral blood (0.23) was much lower than that in the lymph node (0.94) (Supplementary Fig. 17). Similarly, PVL in the peripheral blood was only 0.17%, indicating a negligible migration of malignant cells from the lymph node to the peripheral blood. Therefore, these results underline the need to utilize appropriate clinical material for clonality analysis in HTLV-1-infected individuals, especially those with lymphoma or skin cancer subtypes of ATL.

In conclusion, we introduced the RAISING method employing Sanger sequencing as a cost-effective alternative to HTS analysis

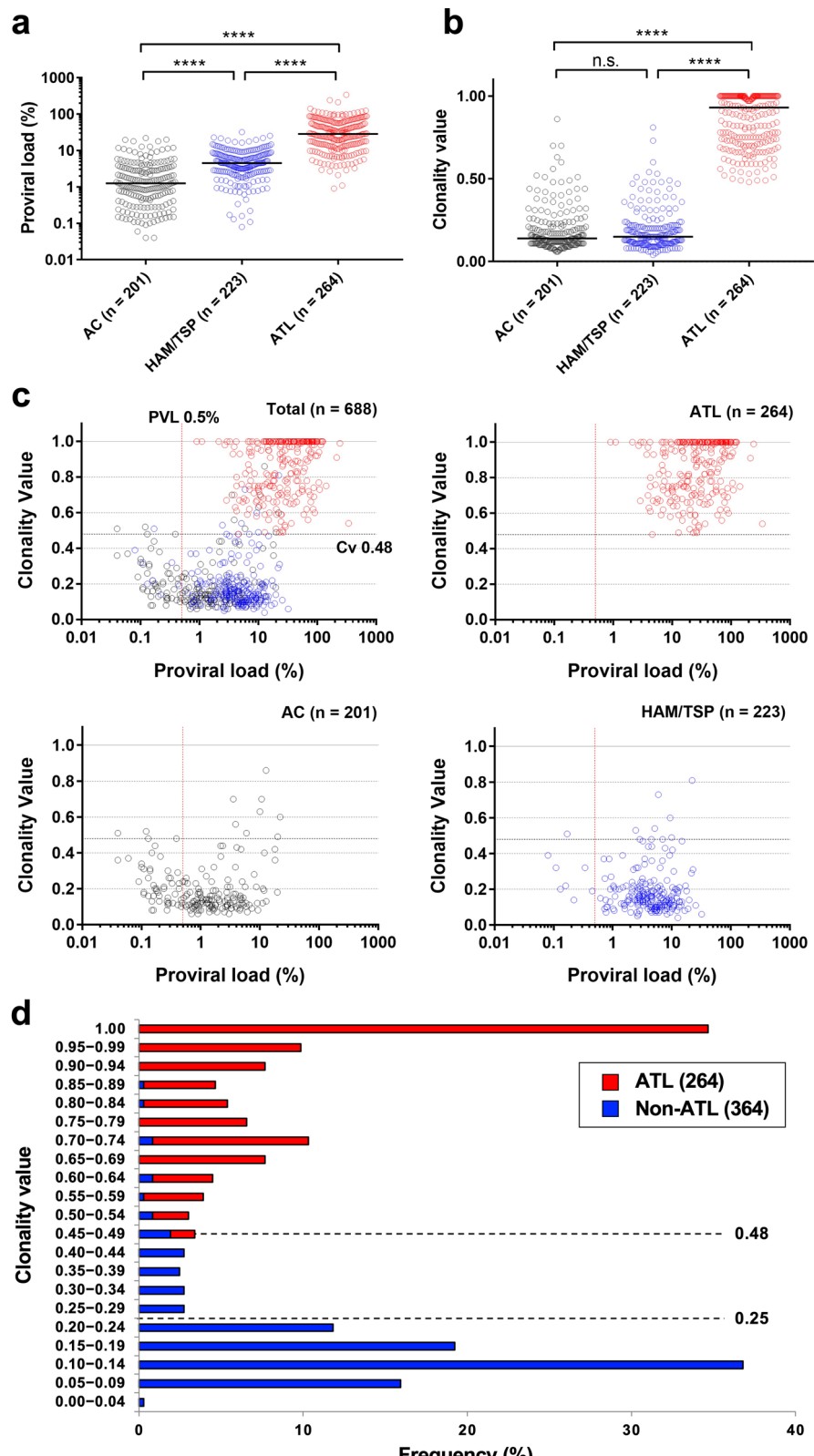

for identifying multiple transgene integration sites. We demonstrated that RAISING is even faster and more sensitive than other previously published methods, and we also introduced a new CLOVA software to facilitate the clonality analysis. Here we focused on HTLV-1 as a model system, showing we could accurately quantify the clonality of infected samples, laying the foundation for using this method in routine clinical testing for HTLV-1-related cancers. Our success measuring the clonality of samples infected with not only HTLV-1 but also BLV, HIV-1, and SIV suggests that RAISING shows promise as a broadly applicable technology. Future studies should explore the full breadth of these applications, especially the use of RAISING to assess the safety and off-target effects of forthcoming gene therapies.

**Fig. 3 Clinical utility of clonality analysis using Rapid Amplification of Integration Site without Interference by Genomic DNA contamination (RAISING) and Clonality Value (CLOVA) software.** HTLV-1 proviral load (PVL, **a**) and clonality values (Cv, **b**) of peripheral blood samples from asymptomatic carriers (AC, $n = 201$, black), HTLV-1-associated myelopathy/tropical spastic paraparesis (HAM/TSP, $n = 223$, blue), and adult T-cell leukemia/lymphoma (ATL, $n = 264$, red) patients. The median PVL and Cv in each group and subtype are shown as horizontal lines. $p$-value were calculated using Dunn's multiple comparisons test. ****$p < 0.0001$, n.s., not significant. **c** Combinational analysis of PVL and Cv using the same samples as above. Cv 0.48 indicate proposed cut-off values to differentiate ATL from non-ATL. PVL 0.5% (red line) or greater means that Cv is accurate. **d** Frequencies of subjects with ATL (red) and without ATL (blue) per Cv at intervals of 0.05. Dotted lines show proposed cut-off values to classify HTLV-1-infected patients into three potential ATL risk zones.

**Table. 1 Classification of a potential Adult T-cell leukemia/lymphoma (ATL) risk based on clonality value of HTLV-1-infected cells.**

| Zone[a] | PVL (%) | Cv | ~ First clone size (%) | Non-ATL ($n = 424$) | | ATL ($n = 264$) (%) |
| --- | --- | --- | --- | --- | --- | --- |
| | | | | AC ($n = 201$) (%) | HAM/TSP ($n = 223$) (%) | |
| 1 | <0.5[b] | NA | <~13 | 88.6 | 84.3 | 0.0 |
| | ≥0.5[b] | <0.25 | | | | |
| 2 | ≥0.5[b] | 0.25–0.47 | ~13–40 | 7.5 | 11.7 | 0.0 |
| 3 (ATL zone) | ≥0.5[b] | ≥0.48 | ≥~40 | 4.0 | 4.0 | 100 |

*Cv* clonality value, *PVL* proviral load, *NA* not applicable, *AC* asymptomatic carrier, *HAM/TSP* HTLV-1-associated myelopathy/tropical spastic paraparesis.
[a]This method classifies non-ATL subjects into three zones using samples from peripheral blood.
Zone 1: a typical non-ATL.
Zone 2: a potential progression to ATL.
Zone 3: a potential ATL complication.
[b]Rapid Amplification of Integration Site without Interference by Genomic DNA contamination (RAISING)-CLOVA can reliably measure Cv when PVL is at least 0.5%, assuring the accuracy of this method.

## Methods

**Collection of human and animal samples.** Peripheral blood and biopsies (skin, lymph nodes, and bone marrow) from asymptomatic carriers, HAM/TSP, and ATL patients were harvested after obtaining informed consent at Nagasaki University Hospital, St. Marianna University School of Medicine, Oita University, University of Miyazaki, and as a collaborative project of the Joint Study on Prognostic Factors of ATL Development (JSPFAD). Regarding ATL patients in this study, the Shimoyama classification was used to diagnose and classify the subtypes[35]. Of 264 total subjects in Fig. 3, at least 170 subjects confirmed that Southern blotting analysis was conducted as the confirmatory test[13,19]. This study was primarily aimed at measuring HTLV-1 clonality in peripheral blood using our technology, RAISING. Thus, ATL patients who had or might have a clonal expansion of the malignant cells only in the other tissues were excluded in this study unless there were available biopsies. This study was approved by the research ethics committee of Oita University (198), University of Miyazaki [972(G)], Nagasaki University (16072504), The University of Tokyo (17-118), St. Marianna University School of Medicine (1646), and National Institute of Infectious Diseases (1120).

Peripheral blood and lymph nodes were harvested from an aleukemic (Holstein breed, female, 1 year old) and a leukemic BLV-infected cattle (Holstein breed, female, 4 years old) at dairy farms in Japan. All experimental procedures were conducted following approval from the local committee for animal studies of Hokkaido University (17-0024). Collection of BLV-infected cattle specimens was approved as a simple general permission procedure for using blood samples for assays. Verbal informed consent was obtained from all animal owners.

**Cell culture.** The HTLV-1-infected cell lines TL-Om1, LMY2, ED, KK1, and SLB-1; the HTLV-1-negative acute T-cell leukemia cell line Jurkat (Clone E61: ATCC TIB152); the CD4$^+$ human T-cell line PM1 (3038, NIH AIDS Reagent Program); and the cynomolgus macaque T-cell line HSC-F (JCRB1164) were cultured in RPMI 1640 medium supplemented with 10% fetal bovine serum, 100 U/mL penicillin, and 100 μg/mL streptomycin at 37 °C in a 5% $CO_2$ atmosphere[27,28,36–38]. The adenovirus-infected HEK-293 cell line (JCRB9068) was cultured in Dulbecco's modified Eagle medium supplemented with 10% fetal bovine serum at 37 °C in a 5% $CO_2$ atmosphere[39].

**In vitro HIV-1 and SIV infection.** HIV-1 89.6 was produced by transfection of the p89.6 plasmid into 293T cells using Lipofectamine 2000 (Thermo Fisher Scientific, Waltham, MA, USA) according to the manufacturer's protocol. The cell culture supernatant was used to transduce the virus into PM1 cells, and then the infected cells were harvested at day 7 post-infection. HSC-F cells were infected with SIV-mac293 in viral solution at a multiplicity of infection (MOI) of 0.005 plaque-forming units per cell, and the infected cells were harvested at day 6 post-infection.

**Genome editing.** *Ldlr*-knock-in mice were generated using a genome-editing technology as previously described[40]. The mouse embryos at the pronuclear stage were obtained using in vitro fertilization, and 100 ng/μL of *Cas9* mRNA, 50 ng/μL

of trans-activating crRNA (tracrRNA), 25 ng/μL of two CRISPR RNAs (crRNAs), and 50 ng/μL of long single-stranded DNA (lssDNA) were simultaneously microinjected into the cytoplasm of embryos. After culturing in KSOM medium (Merck, Darmstadt, Germany) overnight, the embryos dividing into two cells were transferred into pseudo-pregnant females. The tracrRNA, crRNA, and lssDNA were synthesized by FASMAC Co., Ltd.

**Preparation of genomic DNA.** Genomic DNA isolated from PBMCs and biopsies of human specimens and from the TL-Om1, LMY2, ED, Jurkat, and PM1 cell lines was purified using a QIAamp DNA Blood Mini kit (Qiagen, Hilden, Germany). At the St. Marianna University School of Medicine, genomic DNA was extracted from PBMCs and WBCs using overnight proteinase K digestion followed by phenol/chloroform extraction. In contrast, genomic DNA of whole blood cells was purified using Quick-DNA Miniprep Kit (Zymo Research, Irvine, CA, USA). Genomic DNA isolated from PBMCs and lymph nodes of cattle specimens was purified using a Wizard Genomic DNA Purification kit (Promega, Madison, WI, USA). Genomic DNA isolated from the HSC-F cell line was purified using a DNeasy Blood & Tissue Kit (Qiagen, Hilden, Germany). Genomic DNA isolated from the HEK-293 cell line was purified using Isogen-LS (Nippon Gene, Tokyo, Japan). Genomic DNA of KK1 and SLB-1 was extracted using NEBNext Single Cell Lysis Module with Thermolabile Proteinase K and RNase A (New England BioLabs, Ipswich, MA, USA).

**Proviral load (PVL) analysis.** Quantitative polymerase chain reaction (qPCR) for PVL of HTLV-1 was performed using LC480 (NIPPON Genetics, Tokyo, Japan) with the following primer sets for HTLV-1: probe, 5′-CCAGTCTACGTGTTTG-GAGACTGTGTACA-3′; forward primer, 5′-CCCACTTCCCAGGGTTTGGA-3′; reverse primer, 5′-GGCCAGTAGGGCGTGA-3′. β-globin was used as an internal control and amplified with the following primer sets: probe, 5′-AAGGT-GAACGTGGATGAAGTTGGTGG-3′; forward primer, 5′-GTGCACCT-GACTCCTGAGGAGA-3′; reverse primer, 5′-CCTTGATACCAACCTGCCCAG-3′. At the St. Marianna University School of Medicine, PVL was measured using qPCR as previously described[41], and was standardized using the relative ratio determined in a previous study[42]. The qPCR for BLV provirus was performed using Cycleave PCR Reaction Mix (Takara Bio, Otsu, Japan) and Probe/Primer Mix for BLV (Takara Bio) with a LightCycler 480 system II (Roche Diagnostics, Mannheim, Germany). Serial dilution of a BLV positive control (Takara Bio) was used to generate calibration curves to determine the copy number of the BLV *tax* gene. Each DNA sample was tested in duplicate. The concentration of DNA was estimated by measuring the ultraviolet absorbance at 260 nm using a NanoDrop 8000 spectrophotometer (Thermo Fisher Scientific). The reported values are the mean numbers of copies per 50 ng of DNA.

**The cell-direct method.** The indicated cell number of KK1 and SLB-1 were suspended in 1 μL of phosphate buffer, and then mixed with 0.5 μL of 10× NEBNext Cell Lysis Buffer (New England BioLabs), 0.5 μL of RNase A (100 ng/μL, New

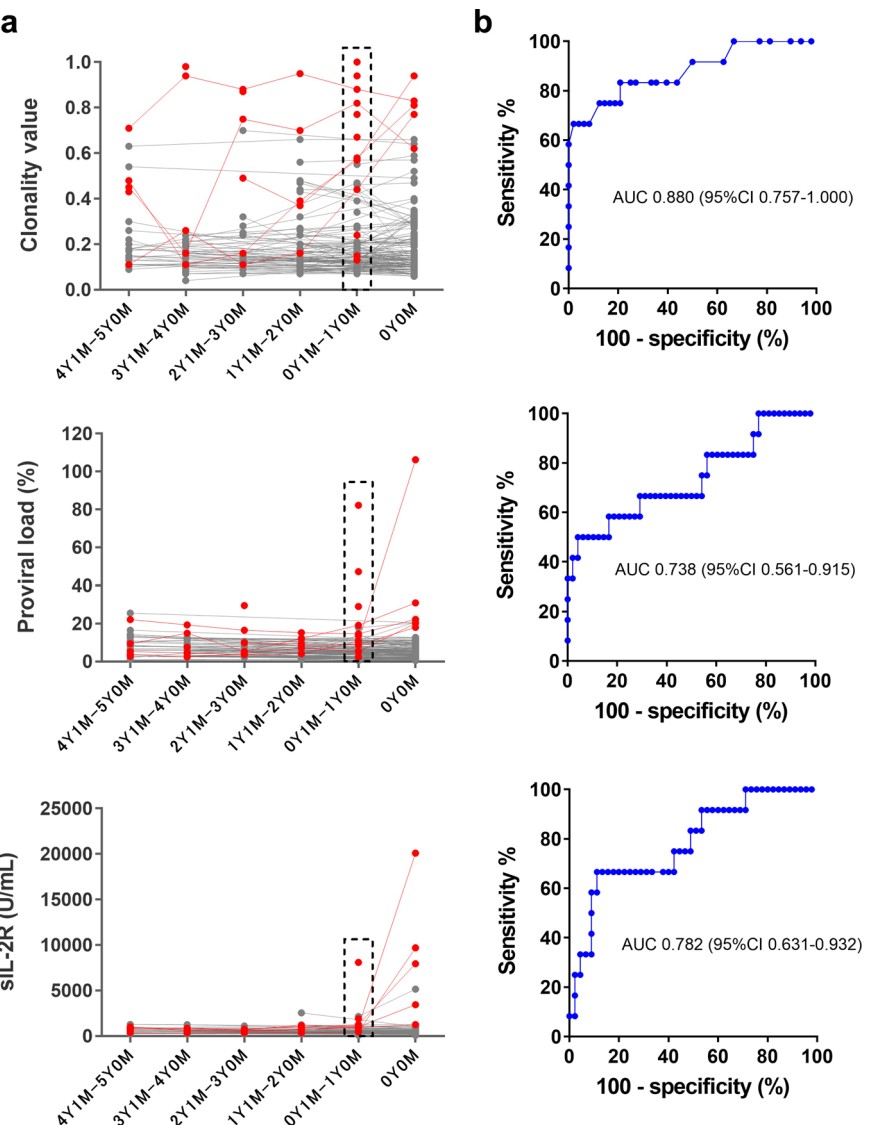

**Fig. 4 Longitudinal analysis of progressors to ATL and non-progressors with clonality value (Cv), HTLV-1 proviral load (PVL), and soluble IL-2 receptor (sIL-2R). a** Cv (top), PVL (center), and sIL-2R (bottom) of 15 progressors (red) and 130 non-progressors (black). The time point of ATL onset or last observation visit was set to 0 year 0 month (0Y0M) for progressors or non-progressors, respectively. If multiple data existed in each period of one year, the arithmetic means were displayed. Of 15 progressors, 10 had only data at one-time point before the onset. Data used for the ROC analysis shown in **b** is indicated by dashed rectangles. **b** ROC analysis using available data from progressors ($n = 12$) and non-progressors ($n = 48$) in a time point between one month and one year (0Y1M–1Y0M) before either ATL onset or last observation visit (progressors, median 7.0 months ago; non-progressor, median 6.2 months ago). AUC area under the curve, CI confidence interval.

| Table. 2 Risk assessment of adult T-cell leukemia/lymphoma (ATL) development. | | | | | | | |
|---|---|---|---|---|---|---|---|
| | Cv | PVL | sIL-2R | | Cv | PVL | sIL-2R |
| Cut-off values[a] for differentiation between middle- and high-risk groups | 0.50 | 13.5 | 1260 | Cut-off values[a] for differentiation between low- and middle-risk groups | 0.24 | 5.0 | 479 |
| Sensitivity (%) | 66.7 | 50.0 | 33.3 | Sensitivity (%) | 83.3 | 83.3 | 83.3 |
| Specificity (%) | 95.8 | 95.8 | 95.6 | Specificity (%) | 79.2 | 43.8 | 51.1 |

*Cv* clonality value, *PVL* proviral load, *sIL-2R* soluble interleukin-2 receptor.
[a]The cut-off values were determined by the ROC analysis shown in Fig. 4b.

England BioLabs), 0.5 µL of Thermolabile Proteinase K (New England BioLabs), and 2.5 µL of distilled water. The mixture was incubated 37 °C for 45 min, then at 55 °C for 10 min. Total 5 µL of the mixture was used for RAISING.

**Sanger sequencing analysis.** Sanger sequencing was performed using the BigDye Terminator v3.1 Cycle Sequencing Kit according to the manufacturer's instructions

(Thermo Fisher Scientific), and the analysis was performed on a 3730Xl DNA Analyzer (Thermo Fisher Scientific).

**Development and application of CLOVA.** We modified EditR[43] to develop CLOVA, an R program-based software that automatically enables analysis of the clonality value (Cv) of transgene-integrated cells, by uploading an "ab.1 file," a

Sanger sequencing file containing the integration site in the host genome sequence, and entering a transgene sequence (up to 20 nucleotides) adjacent to the host genome sequence. The analysis can be performed on a web browser with the R Shiny package[44], but without any cumbersome command-line operations. Methods to obtain the QC filtering, total peak area before filtering plot, and percent noise peak in EditR were not modified in CLOVA. CLOVA can display the Cv values and averages of signal peak area values of the transgene and host genome sequences on "Data QA: Signal and noise plot (signal: peak area of a representative nucleotide, noise: total peak area of the other three nucleotides)." As the Cv values and averages of signal peak area values depend on the length of the transgene sequence, the length is alterable in the "bp-length (default value: 20)" of CLOVA corresponding to multiple transgenes. To obtain a stable Cv, we recommend examining the length of the transgene before performing the clonality analysis as we previously described for the analysis of HTLV-1 clonality[19]. CLOVA has an additional function that can output the Sanger sequencing results of the base call (first clone) and second call (second clone) with FASTA format on the "Download FASTA/CSV" tab using sangerseqR[45]. Because the complexity of Sanger sequence spectra patterns and the quality of Sanger sequencing results for the first and second clones depend on the individual clone size, we also recommend adjusting the "threshold for second call (default value: 85)" for the better discrimination of the two clones. CLOVA consists of R shiny package (tested by version 1.5.0)[44], R sangerseqR package (tested by version 1.22.0)[44], R magrittr package (tested by version 1.5)[46], R dplyr package (tested by version 1.0.2)[47], R tidyr package (tested by version 1.1.2.9000)[48], and R plotly package (tested by version 4.9.2.1)[49].

**Quantification of HTLV-1 clonality**. HTLV-1 clonality was quantified as previously described[19], and CLOVA was used for the quantification instead of EditR in this study.

**High throughput sequencing (HTS)**. The HTS library was prepared by performing tailed-PCR using an amplicon obtained in step 6 of RAISING, and the PCR products were purified using the Agencourt AMPure XP kit (Beckman Coulter, Brea, CA, USA). Then, the DNA was quantified using a Qubit dsDNA HS assay kit (Thermo Fisher Scientific) and an Agilent BioAnalyzer with High-Sensitivity DNA chips (Agilent Technologies, Santa Clara, CA, USA). HTS was performed with a MiSeq Reagent Kit v3 (600-cycle) on an Illumina MiSeq system (Illumina, Hayward, CA, USA) according to the manufacturer's instructions.

**HTS data analysis**. After excluding sequence reads shorter than 50 nucleotides, homology searches were performed using Magic-BLAST (https://ncbi.github.io/magicblast/)[50]. We extracted only sequence reads that had both an appropriate viral sequence length (40, 50, 60, and 70 nucleotides for HTLV-1, BLV, HIV-1, and SIV, respectively) and high homology with the host genome sequence (≥95% match score). Viral integration sites were positioned in the *Homo sapiens* (*Homo sapiens* GRCh38.p12), *Bos taurus* (*Bos taurus* ARS-UCD1.2), or *Macaca fascicularis* (Macaca_fascicularis_5.0) reference genomes. On- and off-targets in the *Ldlr*-mLO-4 and *Ldlr*-mLO-5 knock-in mice were identified by homology searches with the mouse reference genome (*Mus musculus* GRCm38.p6).

**Statistics and reproducibility**. Kruskal–Wallis tests followed by Dunn's post-hoc tests were used for comparison among the three groups (AC, HAM/TSP, and ATL) and for comparison among the another three groups (ATL subtypes excluding lymphoma subtype). Receiver operating characteristic (ROC) curve analysis was performed to assess the ability of the HTLV-1 proviral load and clonality value to distinguish between ATL patients and non-ATL HTLV-1-infected individuals (AC and HAM/TSP). Statistical analyses and graph construction were performed using GraphPad Prism 7 (GraphPad Software, Inc., San Diego, CA, USA). Samples sizes and definitions of replicates are reported in the figure legends.

**Longitudinal analysis**. Clinical samples used for this analysis are shown in Supplementary Data 4. Clonality value (Cv) and proviral load (PVL) were measured using a DNA sample of each time point before ATL onset. The concentration of soluble interleukin-2 receptor (sIL-2R) in serum was determined by chemiluminescent enzyme immunoassay (Fujirebio, Tokyo, Japan). ROC analysis was performed to examine abilities of Cv, PVL, and sIL-2R to distinguish between ATL progressors and non-progressors. Cut-off values for assessing the risk of ATL development were determined using ROC analysis. In this regard, the cut-off values showing a 95% or higher specificity with the highest sensitivity were adopted to distinguish between the ATL high-risk and medium-risk groups. Also, the cut-off values showing a sensitivity of 80% or higher with the highest specificity were adopted to distinguish between the ATL low-risk and medium-risk groups.

## Data availability

The high throughput sequence data are available from the DDBJ/EMBL/NCBI Sequence Read Archives under the accession numbers DRA013179. Uncropped gel images are provided in the supplementary information (Supplementary Fig. 18). All other data are available from the corresponding author on reasonable request.

## Code availability

CLOVA is available from the following URL: http://fasmac.co.jp/rais_method_case (Japanese website) and http://fasmac.co.jp/en/rais (English website).

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

# ARTICLE

25. Iwanaga, M. et al. Human T-cell leukemia virus type I (HTLV-1) proviral load and disease progression in asymptomatic HTLV-1 carriers: a nationwide prospective study in Japan. *Blood* **116**, 1211–1219 (2010).

26. Rowan, A. G. et al. Evolution of retrovirus-infected premalignant T-cell clones prior to adult T-cell leukemia/lymphoma diagnosis. *Blood* **135**, 2023–2032 (2020).

27. Hasegawa, H. et al. Sensitivity of adult T-cell leukaemia lymphoma cells to tumour necrosis factor-related apoptosis-inducing ligand. *Br. J. Haematol.* **128**, 253–265 (2005).

28. Maeda, M. et al. Origin of human T-lymphotrophic virus I-positive T cell lines in adult T cell leukemia. Analysis of T cell receptor gene rearrangement. *J. Exp. Med.* **162**, 2169–2174 (1985).

29. Demontis, M. A., Hilburn, S. & Taylor, G. P. Human T cell lymphotropic virus type 1 viral load variability and long-term trends in asymptomatic carriers and in patients with human T cell lymphotropic virus type 1-related diseases. *AIDS Res. Hum. Retroviruses* **29**, 359–364 (2013).

30. Nagai, M. et al. Analysis of HTLV-I proviral load in 202 HAM/TSP patients and 243 asymptomatic HTLV-I carriers: high proviral load strongly predisposes to HAM/TSP. *J. Neurovirol.* **4**, 586–593 (1998).

31. Vine, A. M. et al. Polygenic control of human T lymphotropic virus type I (HTLV-I) provirus load and the risk of HTLV-I-associated myelopathy/tropical spastic paraparesis. *J. Infect. Dis.* **186**, 932–939 (2002).

32. Watanabe, T. Adult T-cell leukemia: molecular basis for clonal expansion and transformation of HTLV-1-infected T cells. *Blood* **129**, 1071–1081 (2017).

33. Marcais, A. et al. Targeted deep sequencing reveals clonal and subclonal mutational signatures in Adult T-cell leukemia/lymphoma and defines an unfavorable indolent subtype. *Leukemia.* **35**, 764–776 (2020).

34. Kataoka, K. et al. Integrated molecular analysis of adult T cell leukemia/lymphoma. *Nat. Genet.* **47**, 1304–1315 (2015).

35. Shimoyama, M. Diagnostic criteria and classification of clinical subtypes of adult T-cell leukaemia-lymphoma. A report from the Lymphoma Study Group (1984–87). *Br. J. Haematol.* **79**, 428–437 (1991).

36. Akari, H. et al. Effects of SIVmac infection on peripheral blood CD4+CD8+ T lymphocytes in cynomolgus macaques. *Clin. Immunol.* **91**, 321–329 (1999).

37. Kuramitsu, M. et al. Identification of TL-Om1, an adult T-cell leukemia (ATL) cell line, as reference material for quantitative PCR for human T-lymphotropic virus 1. *J. Clin. Microbiol.* **53**, 587–596 (2015).

38. Lusso, P. et al. Growth of macrophage-tropic and primary human immunodeficiency virus type 1 (HIV-1) isolates in a unique CD4+ T-cell clone (PM1): failure to downregulate CD4 and to interfere with cell-line-tropic HIV-1. *J. Virol.* **69**, 3712–3720 (1995).

39. Graham, F. L., Smiley, J., Russell, W. C. & Nairn, R. Characteristics of a human cell line transformed by DNA from human adenovirus type 5. *J. Gen. Virol.* **36**, 59–74 (1977).

40. Miyasaka, Y. et al. CLICK: one-step generation of conditional knockout mice. *BMC Genom.* **19**, 318 (2018).

41. Yamano, Y. et al. Correlation of human T-cell lymphotropic virus type 1 (HTLV-1) mRNA with proviral DNA load, virus-specific CD8(+) T cells, and disease severity in HTLV-1-associated myelopathy (HAM/TSP). *Blood* **99**, 88–94 (2002).

42. Kuramitsu, M. et al. Standardization of quantitative PCR for human T-cell leukemia virus Type 1 in Japan: a Collaborative Study. *J. Clin. Microbiol.* **53**, 3485–3491 (2015).

43. Kluesner, M. G. et al. EditR: a method to quantify base editing from Sanger sequencing. *CRISPR J.* **1**, 239–250 (2018).

44. Chang, W. et al. shiny: Web application framework for R. https://CRAN.R-project.org/package=shiny (2014).

45. Hill, J. T. et al. Poly peak parser: method and software for identification of unknown indels using sanger sequencing of polymerase chain reaction products. *Dev. Dyn.* **243**, 1632–1636 (2014).

46. Stefan, M. B. et al. magrittr: a forward-pipe operator for R. https://cran.r-project.org/web/packages/magrittr/index.html (2014).

47. Hadley, W. et al. dplyr: a grammar of data manipulation. https://cran.r-project.org/web/packages/dplyr/index.html (2020).

48. Hadley, W. et al. tidyr: Tidy Messy data. https://cran.r-project.org/web/packages/tidyr/index.html (2020).

49. Carson, S. et al. plotly: Create Interactive Web Graphics via 'plotly.js' https://cran.r-project.org/web/packages/plotly/index.html (2020).

50. Boratyn, G. M. et al. Magic-BLAST, an accurate RNA-seq aligner for long and short reads. *BMC Bioinforma.* **20**, 405 (2019).

## Acknowledgements

We are grateful to Dr. Masao Matsuoka for providing the ED cell line; Dr. Masahiro Fujii for providing the SLB-1 cell line; Drs. Kousho Wakae for providing genomic DNA from the HBV-infected cell line (HepAD38); Dr. Shigeyoshi Harada for providing genomic DNA from HIV-1-infected cells (HIV-1 89.6-infected PM1 cells) and the infection method; Drs. Takushi Nomura, Hirofumi Akari, and Tetsuro Matano for providing genomic DNA from the SIV-infected cell line (SIVmac239-infected HSC-F cells) and the infection method; Mrs. Takao Tanaka and Yoshihiro Ohguchi (KAC Co., Ltd) for providing genomic DNA from knock-in mice (Ldlr-mLO-4 and Ldlr-mLO-5) and the established genome-editing method; and New England Biolabs Japan Inc. for providing OneTaq, NEBNext® Single Cell Lysis Module, and Thermolabile Proteinase K as samples. We also thank the patients who participated in this research as well as the faculty members of the Division of Hematology and Oncology at St. Marianna University School of Medicine (Yu Uemura and Ayako Arai), and laboratory support staff of the Institute of Medical Science at St. Marianna University School of Medicine (Katsunori Takahashi, Yasuo Kunitomo, Yumiko Hasegawa, Mikako Koike, Satoko Aratani, Chihiro Sasa, Yumi Saito, and Miho Ishikawa). This work was supported by grants from JSPS KAKENHI (JP17H03594 and JP 16H06277 CoBiA: M.S., JP18H02733: H.H., JP19H03575: Y.Y., and JP19K07983: T.S.), AMED (JP20ek0109346 and JP20ek0109356: Y.Y.), and Metabolic Skeletal Disorders Training Grant through the National Institute of Arthritis and Musculoskeletal and Skin Diseases (T32AR060719: A.L.G.C.-R.).

## Author contributions

Y.W., T.S., H.H., T.M., S.Y., H.M., M.K., D.S., N.M., and M.S. performed the research; N.N., T.T., H.M., M.T., and K.T. performed the data analysis; T.S., H.H., S.Y., T.O., N.Y., J.Y., N.A., M.Y., M.N., S.N., H.I., M.O., Y.I., K.U., Y.M., S.K., K.Y., K.M., T.W., and Y.Y. provided patient samples; Y.W., T.S., H.H., T.M., T.T., A.L.G.C.-R., Y.Y., and M.S. wrote the manuscript; and T.S., H.H., M.M., Y.Y., and M.S. supervised the study.

## Competing interests

Y.W., T.M., H.M., and M.S. are inventors on the pending patent application (PCT/JP2020/03907) for RAISING technology. This study was partially funded by FASMAC Co., Ltd. The funder provided support in the form of salaries for authors (Y.W. and T.M.) and partial research materials, but did not have any additional role such as the study design, data collection and analysis, decision to publish and preparation of the manuscript. All other authors declare no competing interests.
