## [Peer Review File · Communications Biology]

Reviewers' comments:

Reviewer #1 (Remarks to the Author):

The identification and monitoring of expanded clones, in the case of virally (or viral vector) induced malignancy is important for clinical and research purposes. While low-cost methods (such as IPCR) exist, and are used clinically around the world, they also lack sensitivity and are not quantitative, making their use for monitoring limited. Newer methods are sensitive and quantitative but can be prohibitively expensive for clinical or research use in many parts of the world. Here the authors present a novel method, RAISING, which they report would provide (alongside their software solution CLOVA) an accessible, affordable, and relatively rapid solution.

Major points:

1. The key message in the manuscript (the discussion of a new method) is unclear, due to the discussion of a previous method, RAIS. Authors should focus discussion of their method to the new method itself, after the QC.
2. The results from the 3 ATL patients are interesting but are insufficient to propose a prognostic or predictive value due to the low number and due to the lack of appropriate controls.
3. Based on the methods (line 414), ATL patients were selected based on monoclonality – this may call the analysis shown in figure 3C and the ROC analysis into question.

Minor points:

1. The primer naming strategy is confusing, with two different primers called F1.
2. Line 254-257 – the mention of a software not used is not necessary (the reference in the methods, given that CLOVA is based on EditR, is however justified).
3. How well does the CV index correlate with the OCI (mentioned in line 318)?
4. Line 534 – as previously described where?
5. Supplementary fig. 1a – is it possible that the panels were mislabeled? Not clear the discrepancy between left and right panels.
6. Authors should clarify what they mean by contaminated genomic DNA (e.g., line 167).
7. In NGS analysis - if using read counts to determine clonality, can the authors distinguish between preferential amplification of specific PCR templates and clonal expansion (e.g. UMI)?
8. The method relies on the 3'LTR for detection of integration, however authors mention that in some cases the malignant clone is absent the 3'LTR. The authors can discuss potential approaches for how their method could be expanded to include both LTRs.
9. CLOVA application - The large number of dependencies is a concern as their behavior/availability over time may change. Authors should clarify which version of the dependencies has been tested with their application and where possible remove dependencies.
10. Authors should explain the meaning of the 'noise' and 'signal' in CLOVA – how are those defined/distinguished?

Reviewer #2 (Remarks to the Author):

In the present study reported by Y. Wada et al., authors presented a novel method of tracking transgene integration sites called RAISING. This method follows a previous method called RAIS. First, authors describe extensively the technique, which provides several improvements compared to the previous method.

The authors take advantage of an impressive collection of HTLV-1 samples which encompass all clinical forms of HTLV-1 infected patients (ATL, TSP, HC).

The first part of the manuscript describes extensively and very precisely the technique. The second part analyzes its efficacy which has a very good sensibility as it can amplify integrated fragments in samples with a proviral load (PvL) less than 0.032%. Authors show also that the technique can be followed by NGS analysis as well as Sanger sequencing and can analyses various transgene (BLV, HIV...). The authors develop an algorithm that provides a clonality value (CV) and demonstrate that, paired with PvL analysis, it can delineate ATL from non ATL patients.

Finally, authors analyzed longitudinal samples from 3 HC/TSP patients who developed Acute ATL. Results seems to show that the increase of the CV conversely to PvL, can anticipate the onset of

ATL.

The paper is well written and the scientific procedures are well described. The authors take advantage of a unique collection of well annotated samples and improve on their already excellent previous contributions to the field. This is a very interesting work on a topic that has progressively gained pathophysiological insights in recent years. However, it would require further studies to assess the impact of this method on the clinical management of HTLV1/ATL patients.

Comments

1. In figure 3a and b and sup figure 13. Authors analyzed very few lymphoma subtypes (n=10) compared to other ATL subtypes. How authors justify it? Moreover, in lymphoma subtype, there is no tumoral cells according to the Shimoyama classification. How authors explained that the CV is at the same range as ATL leukemic subtypes?
2. The longitudinal study seems interesting and promising but included only 3 patients compared to the collection of more than 600 samples at diagnosis. It would be helpful to analyze more longitudinal samples to confirm these results and also longitudinal samples of HC/TSP patients who do not develop ATL.

December 24, 2021
Dr. Brooke LaFlamme
Editor-in-Chief
Communications Biology

Dear Dr. Brooke LaFlamme:

Thank you for your review and response regarding our manuscript, "**RAISING: a high-performance method for identifying random transgene integration sites**," for consideration of publication as an Article in *Communications Biology* (COMMSBIO-21-1255A).

We have carefully read the reviewer's comments and responded to each in blue font. Our manuscript has been precisely revised according to the comments.

Reviewers' comments:

Reviewer #1 (Remarks to the Author):

The identification and monitoring of expanded clones, in the case of virally (or viral vector) induced malignancy is important for clinical and research purposes. While low-cost methods (such as IPCR) exist, and are used clinically around the world, they also lack sensitivity and are not quantitative, making their use for monitoring limited. Newer methods are sensitive and quantitative but can be prohibitively expensive for clinical or research use in many parts of the world. Here the authors present a novel method, RAISING, which they report would provide (alongside their software solution CLOVA) an accessible, affordable, and relatively rapid solution.

Major points:

1. The key message in the manuscript (the discussion of a new method) is unclear, due to the discussion of a previous method, RAIS. Authors should focus discussion of their method to the new method itself, after the QC.

Our response: We have excluded sentences and Supplementary Fig.11 regarding RAIS after QC in the initially submitted manuscript, and focused on RAISING in the revised manuscript according to the reviewer's comment.

2. The results from the 3 ATL patients are interesting but are insufficient to propose a prognostic or predictive value due to the low number and due to the lack of appropriate controls.

Our response: In the revised manuscript, we have performed an additional longitudinal study using samples of 15 progressors to ATL and 130 non-progressors (26 asymptomatic carriers and 104 HAM/TSP patients) according to the reviewer's comment (Fig.4 and supplementary Table 4 in the

revised manuscript). The ROC analysis showed that clonality value was the most significant prediction marker of ATL onset.

3. Based on the methods (line 414), ATL patients were selected based on monoclonality – this may call the analysis shown in figure 3C and the ROC analysis into question.

Our response: We mentioned “clonal band (s)” in the initially submitted manuscript. This did not mean only “monoclonality” as the reviewer claimed. However, we understood the reviewer #1’s concern and recognized that the sentence was incorrect. Thus, we have precisely revised the sentences describing how ATL patients were diagnosed and how many of the subjects were subjected to Southern blotting analysis to confirm the diagnosis (line 401–407).

Minor points:

1. The primer naming strategy is confusing, with two different primers called F1.

Our response: We have revised the primer name according to the reviewer’s comment (Fig. 1 and Supplementary Table 1).

2. Line 254-257 – the mention of a software not used is not necessary (the reference in the methods, given that CLOVA is based on EditR, is however justified).

Our response: We have revised the sentences according to the reviewer’s comment (line 497).

3. How well does the CV index correlate with the OCI (mentioned in line 318)?

Our response: The correlation between Cv and OCI has been shown in Supplementary Fig. 14 of the revised manuscript according to the reviewer’s comment. We have added the sentences in the revised manuscript (line 303-307).

4. Line 534 – as previously described where?

Our response: We have mentioned the reference in the sentence according to the reviewer’s comment (line 524).

5. Supplementary fig. 1a – is it possible that the panels were mislabeled? Not clear the discrepancy between left and right panels.

Our response: We have revised the sentences according to the reviewer’s comment (Supplementary Fig.1 legend).

6. Authors should clarify what they mean by contaminated genomic DNA (e.g., line 167).

Our response: We have mentioned the meaning of “contaminated genomic DNA” in the revised manuscript according to the reviewer’s comment (line 158–161).

7. In NGS analysis - if using read counts to determine clonality, can the authors distinguish between preferential amplification of specific PCR templates and clonal expansion (e.g. UMI)?

Our response: Using NGS analysis, we have shown RAISING as an unbiased method (Fig.1d). We also have mentioned a limitation that mutations in transgene-specific primer binding sites affect clonality and read counts in NGS analysis (line 367–373). Although all primers were designed in regions where mutation frequency was extremely low, or none, the preferential amplification of specific PCR templates might occur when the primer binding sites in some templates were mutated. Thus, we have just recommended using southern blot analysis or target-capture sequencing (Nagasaka, M, et al, PNAS, 2020) for them in the revised manuscript.

8. The method relies on the 3’LTR for detection of integration, however authors mention that in some cases the malignant clone is absent the 3’LTR. The authors can discuss potential approaches for how their method could be expanded to include both LTRs.

Our response: As previously reported (Tamiya S, et al, Blood, 1996), deletions of HTLV-1 5’LTR was observed in approximately 40% of ATL samples. In addition, we found that the pattern of the 5’LTR deletion varied in each ATL sample (Data not shown) and realized the difficulty of primer design on the HTLV-1 5’LTR region. On the other hand, only 2% of ATL samples showed 3’LTR deletion as described in this study. Thus, we have developed RAISING that relies on the 3’LTR but not the 5’LTR to detect the integration site (Saito, M, et al, IJH, 2020). Although potential approaches for detecting both LTRs may be good to discuss, we have simply recommended using southern blot analysis or target-capture sequencing (Nagasaka, M, et al, PNAS, 2020) as the alternative method in the revised manuscript.

9. CLOVA application - The large number of dependencies is a concern as their behavior/availability over time may change. Authors should clarify which version of the dependencies has been tested with their application and where possible remove dependencies.

Our response: We have clarified and constructed a new sentence regarding the version of the dependencies according to the reviewer’s comment (Method, Development and application of CLOVA, line 517–521).

10. Authors should explain the meaning of the ‘noise’ and ‘signal’ in CLOVA – how are those defined/distinguished?

Our response: We have explained the meaning of the ‘noise’ and ‘signal’ in CLOVA in the revised

manuscript according to the reviewer's comment (line 504–507).

Reviewer #2 (Remarks to the Author):

In the present study reported by Y. Wada et al., authors presented a novel method of tracking transgene integration sites called RAISING. This method follows a previous method called RAIS.

First, authors describe extensively the technique, which provides several improvements compared to the previous method.

The authors take advantage of an impressive collection of HTLV-1 samples which encompass all clinical forms of HTLV-1 infected patients (ATL, TSP, HC).

The first part of the manuscript describes extensively and very precisely the technique. The second part analyzes its efficacy which has a very good sensibility as it can amplify integrated fragments in samples with a proviral load (PVL) less than 0.032%. Authors show also that the technique can be followed by NGS analysis as well as Sanger sequencing and can analyses various transgene (BLV, HIV...). The authors develop an algorithm that provides a clonality value (CV) and demonstrate that, paired with PVL analysis, it can delineate ATL from non ATL patients.

Finally, authors analyzed longitudinal samples from 3 HC/TSP patients who developed Acute ATL. Results seems to show that the increase of the CV conversely to PVL, can anticipate the onset of ATL.

The paper is well written and the scientific procedures are well described. The authors take advantage of a unique collection of well annotated samples and improve on their already excellent previous contributions to the field. This is a very interesting work on a topic that has progressively gained pathophysiological insights in recent years. However, it would require further studies to assess the impact of this method on the clinical management of HTLV1/ATL patients.

Comments

1. In figure 3a and b and sup figure 13. Authors analyzed very few lymphoma subtypes (n=10) compared to other ATL subtypes. How authors justify it? Moreover, in lymphoma subtype, there is no tumoral cells according to the Shimoyama classification. How authors explained that the CV is at the same range as ATL leukemic subtypes?

Our response: We have understood the reviewer's comment. In our study, the clonality value (Cv) of lymphoma subtypes (n=10) was analyzed using peripheral blood samples. The Cv was at the same range as ATL leukemic subtypes. Our results suggest the migration of malignant cells from lymphoid tissues to peripheral blood in lymphoma patients. However, we could not confirm the existence of the same dominant clone (malignant cells) in both peripheral blood and lymphoid tissues (no available material). Thus, we have decided to exclude results of the lymphoma subtype in Fig.3 and Fig.4 of the revised manuscript except Supplementary Fig.17, showing a typical lymphoma type.

2. The longitudinal study seems interesting and promising but included only 3 patients compared to the collection of more than 600 samples at diagnosis. It would be helpful to analyze more longitudinal

samples to confirm these results and also longitudinal samples of HC/TSP patients who do not develop ATL.

Our response: In the revised manuscript, we have performed the additional longitudinal study using samples of 15 progressors to ATL and 130 non-progressors (26 asymptomatic carriers and 104 HAM/TSP patients) according to the reviewer's comment (Fig.4 and supplementary Table 4). The ROC analysis showed that clonality value was the most significant prediction marker of ATL onset.

Other revisions

1. Haruka Momose has been included in "Competing interests statement" in the revised manuscript.
2. Affiliation of Naganori Nao and Masumichi Saito has been changed in the revised manuscript.
3. Nariyoshi Matsumoto has been added in "co-authors" and "Author contributions" in the revised manuscript.

Thank you very much for your reconsideration. I look forward to hearing from you again.

Sincerely,

Corresponding Author: Masumichi Saito, Ph.D.
Department of Safety Research on Blood and Biological Products,
National Institute of Infectious Diseases,
Gakuen 4-7-1, Musashimurayama,
Tokyo 208-0011, Japan
Tel: 81-42-561-0771, Fax: 81-42-565-3315
E-mail: saitomas@nih.go.jp

Co-Corresponding Author: Yoshihisa Yamano, M.D., Ph.D.
Department of Rare Diseases Research, Institute of Medical
Science, St. Marianna University School of Medicine,
2-16-1 Sugao, Miyamae-ku, Kawasaki, Kanagawa 216-8512,
Japan
Tel: 81-44-977-8111, Fax: 81-44-977-9772
E-mail: yyamano@marianna-u.ac.jp

REVIEWERS' COMMENTS:

Reviewer #1 (Remarks to the Author):

The novel method (RAISING) and software solution (CLOVA) presented in this manuscript are welcomed as a flexible, affordable, and relatively rapid solution to the problem of detection and monitoring of clonal integration of viruses and viral vectors. The additional experiments carried out by the authors reinforce the utility of the clonality measure defined in diagnosis and prognosis, at least in the case of leukemogenesis caused by HTLV-1 infection.

The revised manuscript is greatly improved, and the authors satisfied my questions and comments adequately. I believe that at its current form the manuscript will make a good addition to Communications Biology.

Reviewer #2 (Remarks to the Author):

The authors responded to my comments correctly and the revised manuscript is improved well.